# Prevention from Sharp Injuries in the Hospital Sector: An Italian National Observatory on the Implementation of the Council Directive 2010/32/EU before and during the COVID-19 Pandemic

**DOI:** 10.3390/ijerph191711144

**Published:** 2022-09-05

**Authors:** Gabriella De Carli, Alessandro Agresta, Maria Giuseppina Lecce, Patrizia Marchegiano, Gianpaolo Micheloni, Dimitri Sossai, Giuseppe Campo, Paola Tomao, Nicoletta Vonesch, Sara Leone, Vincenzo Puro

**Affiliations:** 1UOC Infezioni Emergenti e Riemergenti e CRAIDS, Dipartimento di Epidemiologia, Ricerca Preclinica e Diagnostica Avanzata, INMI L. Spallanzani-IRCCS, 00149 Rome, Italy; 2Ufficio IV—Direzione Generale della Prevenzione del Ministero della Salute, 00144 Rome, Italy; 3SC Servizio Prevenzione e Protezione Aziendale, Azienda Ospedaliero-Universitaria di Modena, 41124 Modena, Italy; 4Servizio Medicina del Lavoro, ASST Grande Ospedale Metropolitano Niguarda, 20162 Milano, Italy; 5U.O. Servizio Prevenzione e Protezione, Ospedale Policlinico San Martino, 16132 Genova, Italy; 6INAIL—Sezione Sistemi di Sorveglianza e Gestione Integrata del Rischio, Dipartimento di Medicina, Epidemiologia, Igiene del Lavoro ed Ambientale, 00144 Rome, Italy; 7INAIL—Laboratorio Rischio Agenti Biologici, Dipartimento di Medicina, Epidemiologia, Igiene del Lavoro ed Ambientale, 00078 Monte Porzio Catone, Italy; 8UOC Epidemiologia Clinica, Dipartimento di Epidemiologia, Ricerca Preclinica e Diagnostica Avanzata, INMI L. Spallanzani-IRCCS, 00149 Rome, Italy

**Keywords:** health personnel, nurses, sharps injuries, needlestick injuries, accident prevention, bloodborne pathogens, COVID-19, accident prevention/legislation and jurisprudence, safety-engineered devices, vaccination

## Abstract

Sharp injuries, determining the risk of bloodborne infections and psychological distress in healthcare workers, may be prevented by a set of strategies, legally enforced in Europe through the Directive 2010/32/EU. To assess its level of implementation in Italy, a national survey was conducted in 2017 and again in 2021, evaluating the progress and possible drawbacks of the COVID-19 pandemic. Altogether, 285 safety managers and 330 nurses from a representative sample of 97 and 117 public hospitals were interviewed using a standardized questionnaire. Knowledge of the Directive requirements decreased significantly, with <60% of participants answering correctly in 2021, and nurses’ attendance in specific courses dropped to 25% in 2021 compared to 54% in 2017. Over 75% of hospitals introduced multiple safety-engineered devices (SED), though total replacement occurred in <50% of cases; routine SED availability increased for blood collection (89%) and venous access devices (83%). Incorrect behaviors in handling sharps decreased significantly over time. Nurses’ HBV vaccination coverage was high (89% in both surveys); in the last year, 97% were vaccinated against COVID, and 47% against influenza. Average annual injuries per hospital did not increase significantly (32 in 2021 vs. 26 in 2017). In 2017, nurses’ perceived safety barriers were working in emergency situations (49%) and lack of resources (40%); in 2021, understaffing (73%), physical fatigue (62%), and handling difficulties while wearing full protective equipment (59%). Safety measures were implemented in Italian hospitals, and although the average injuries per hospital did not show a decrease, these measures could have helped protect healthcare workers during the pandemic, mitigating its potential impact on the increase in situations at risk of injury.

## 1. Introduction

After a decade from the entry into force, in May 2011, of the Directive 2010/32/EU for the prevention of sharp injuries in the hospital and healthcare sector [1], the European institutions and the Social Partners who collaborated in its development and promotion, are deeply interested in evaluating the level of implementation and impact of the proposed interventions in reducing the risks of exposure and infection with bloodborne pathogens for healthcare professionals in the Member States, also as a possible example for future regulatory action on worker safety.

The preventive interventions included in the Directive derive from the long experience accumulated since the first cases of occupational HIV infection following a needlestick injury in healthcare workers (HCWs) were identified. HIV gave an unprecedented impulse to HCWs’ safety, leading to the promotion of Universal Precautions against bloodborne infections [2] and the development of surveillance programs for occupational injuries. At first, these were aimed at estimating the probability of seroconversion and identifying possible determinants associated with an increased risk of infection [3,4,5]. Afterward, the focus was shifted to the analysis of injury mechanisms [6], to identify possible preventive measures [7], and the study of the effect of behavioral and technical interventions on injury prevention. These studies found that these accidents were complex events, with multiple determinants and possible contributing factors. Consequently, over the last forty years, multiple interventions were implemented to curb sharp injuries, most often in combination [8,9,10,11], determining a reduction in the frequency of sharp injuries connected to specific mechanisms, but with a limited impact on global injury rates. A more significant decrease was determined by the introduction of safety-engineered devices (SED), which integrate a mechanism for needlestick prevention, and therefore, reduce the exposure by modifying or isolating the hazard [12,13]. Although SED effectiveness is variable [7,8], possibly also because of differences in the design of the safety mechanism, either requiring intentional activation by the HCW or self-activated [13], their implementation reduced the risks of injury. A meta-analysis of intervention strategies to prevent sharp injuries found that the summary effect of studies adopting training on safe injection procedures was a decrease of 34% in the injury rate (95% confidence interval, CI, 11–50%); studies using the implementation of SED obtained a 49% decrease (95% CI 36–60%) while combining the two intervention approaches (training and SED) appeared to be more effective than either intervention alone: a 62% decrease was calculated (95% CI 50–72%) [14], supporting the conclusions that successful injury prevention needed to encompass many different interventions [4]. 

However, the causes of sharp injuries are multifactorial and include elements, such as the types of devices and procedures [15,16], professional inexperience [17] or, on the contrary, experienced familiarity [18], lack of training on prevention and management of sharp injuries [19], improper management of sharps [20], poor organizational climate, high workload and fatigue [21]. As a consequence, only a wider and more structured intervention including multiple strategies could try to address this phenomenon adequately and obtain consistent, long-lasting results; its legal enforcement would enhance the likelihood of a widespread implementation [22]. 

Specific legislation issued to reduce percutaneous injuries, the most frequent route of transmission in occupationally acquired infections due to bloodborne pathogens, was approved in the United States at the end of the year 2000 [23], requiring the implementation of a set of interventions (in summary, SED provision, HCWs’ involvement in the selection, frequent revision of exposure-control plans to include advances in technology, and injury recording). The global effect was an immediate drop of about 38% (95% confidence interval, 35 to 41) in 2001 when the NSPA took effect [24]. 

Following the NSPA issue, HCWs in Europe asked for specific legislation to prevent sharp injuries, and the European Commission set up a Sectoral Social Dialogue Committee. The Directive 2010/32/EU was developed through an unprecedented collaboration between HCWs and healthcare employers, beautifully described in the video realized by the European Commission “From needlesticks to sharps: the added value” [25]. The Directive required “to set up an integrated approach establishing policies in risk assessment, risk prevention, training, information, awareness raising and monitoring”, and “to put in place response and follow-up procedures”. Therefore, its entry into force brought along the implementation of a global preventive strategy whose elements are interconnected, and where the impact of each element cannot be estimated separately from others. The overall effect should be to reduce the risk of sharp injuries to healthcare professionals by addressing many of their multifactorial causes.

In Italy, the most significant experience on these risks is that conducted by the network of hospitals participating in the Italian Study on Occupational Risk of HIV and other bloodborne infections (SIROH) [26], coordinated by the National Institute for Infectious Diseases “Lazzaro Spallanzani”—I.R.C.C.S. in Rome. The study initially included only hospitals with an infectious diseases unit, where HIV-infected patients with acute conditions were admitted. Afterward, the SIROH research program was enlarged, and from 1986 to 2018 it collected data on accidental injuries and infection with bloodborne pathogens in about 150 hospitals throughout the country, promoting the introduction of preventive measures and studies on specific aspects of biological risk which were used as a basis in the development of the Directive 2010/32/EU. In June 2013, when the legal transposition of the Directive should have been completed in the Member States, 100 SIROH hospitals participated in a survey to assess the level of implementation of the preventive measures required by the Directive, and their impact on sharp injuries. SIROH representatives actively inspected wards, observed procedures involving sharps use, reported vaccination rates, postexposure protocols and experiences regarding awareness raising, education and training, and eventually shared their own experiences with the other SIROH members during a two-day meeting. Most of the preventive interventions covered by the Directive had been implemented in SIROH hospitals, and an 80% decrease in needlestick injuries was reported; however, it was necessary to invest in the availability and dissemination of safety-engineered devices, the elimination of unnecessary needles, and streamline post-exposure protocols [27]. 

The important results in terms of sharp injury prevention observed in SIROH hospitals following the implementation of the main elements included in the Directive were not generalizable, as most of these highly engaged centers had joined this surveillance and prevention program since the early ‘90s. Therefore, the situation of Italian hospitals needed to be assessed. Legal transposition of Directive 2010/32/EU was completed in Italy in February 2014 [28]. In 2017, the European Biosafety Network (EBN), established in 2010 following the adoption of the Directive2010/32/EU with a commitment to improving the safety of patients and healthcare and non-healthcare workers, proposed to hold its annual summit in Italy and requested national data on the scope and impact of the Directive implementation at a national level. In order to assess the state-of-the-art, a survey was carried out in 2017 in a sample of Italian public hospitals representative by geographical area and number of beds “Italian Observatory on Needle and Sharps Safety for Healthcare Workers”, through a collaboration between SIROH, the Italian Ministry of Health and the National Institute for Insurance against Accidents at Work (Istituto Nazionale per l’Assicurazione contro gli Infortuni sul Lavoro, INAIL). The results of this survey represented a starting point, to be monitored over time (hence the definition of “observatory”). The preliminary results of this survey were presented at the Italian Ministry of Health in October 2017 at the EBN summit.

HCWs’ workload, professional activity and occupational safety were significantly challenged by the appearance of SARS-CoV-2 [29], which disrupted the continuity of the provision of health service and care. In 2021, the EBN commissioned a survey to understand whether, why and how there had been a change in the number, type and location of sharp injuries as a result of the COVID pandemic. Findings show that the number of sharp injuries has risen significantly in 2020, with an average reported increase of 23% over the last year (an estimated increase of 276,000 sharp injuries). European HCWs blamed increased pressure of work and stress due to COVID, a lack of safety devices and of personal protective equipment [30], and the EBN recommended a consistent interpretation and universal implementation of the existing Sharps Directive [31]. 

Hence, the national survey in Italy was repeated in 2021, during the COVID-19 pandemic, “Italian Observatory 2021 on Needle and Sharps Safety and Hazardous Drugs”, in relation to both the application of Directive 32/2010 and the amendments to the Community legislation on carcinogens, mutagens, and reprotoxins at work. Its aims were to evaluate whether the level of implementation of the preventive interventions had increased in comparison to the previous observations, whether there was a sustained effect on the number of injuries reported by hospitals, and to evaluate the possible effects of the COVID-19 pandemic on sharps safety. 

This article presents the results relating to the prevention of sharp injuries in the two surveys, which involved, through direct interviews, safety managers, hospital pharmacists, and nurses from surgery, internal medicine, infectious diseases and oncology, representing the healthcare workers most affected by the preventive measures introduced.

## 2. Materials and Methods

### 2.1. Survey: Structure and Method of Administration of Interviews

Starting from the questionnaire on the implementation of the Directive administered to the SIROH hospitals in June 2013 [27], through the collaboration between a scientific board including SIROH members, a representative from INAIL and the Ministry of Health, and an expert provider in social and market research (GfK Eurisko—2017, and IQVIA Italia—2021), a standardized interview scheme lasting about 30 min was developed, specific according to the category of the interviewee. The scientific board revised the contents and structured the questions order to obtain crosschecks of the statements. 

Regarding sharps safety, in 2017 the standardized questionnaire used for the interview included 79 closed and three open questions, organized into nine sections which followed the elements included in Directive 32/2010/EU: knowledge of the requirements of the Directive; information/awareness raising; education on risks from bloodborne pathogens and infection prevention and training in the use of sharps; behaviors regarding the use and elimination of sharps; procedures for sharps elimination; adoption of devices integrating a safety mechanism (safety-engineered devices); vaccinations; injuries and management protocol; obstacles to achieving sharps safety. The questionnaire was reduced in 2021 as the interview had to include also questions on hazardous drugs; two sections were eliminated (information, and procedures for sharps elimination), and the final version included 40 closed and one open question organized in seven sections, with nine questions targeted according to the job category of the interviewee (safety manager or nurse), and 32 questions in common, including the open question regarding obstacles to achieving optimal levels of safety in the use of sharps. Only the 34 questions which were maintained completely unaltered from the 2017 version were used in the comparisons. 

The interviews were conducted in the period June 2017–January 2018 with CAPI methodology (Computer Assisted Personal Interview) and in the period July–September 2021 with CAWI methodology (Computer Assisted Web Interview). 

The provider ensured randomization of the sample, and anonymization of the interviews, whose data were made available for analysis in an anonymous and aggregated form, with no possibility to identify the interviewee or the hospital directly or indirectly. The study was conducted in accordance with the Declaration of Helsinki [32], and informed consent was obtained from all subjects involved in the study. According to local and national regulations, the study was exempted from formal clearance, however, it notified the Ethics Committee of the National Institute for Infectious Diseases “Lazzaro Spallanzani”—I.R.C.C.S. 

### 2.2. Sample of Hospitals

The research focused on public hospitals with at least 200 beds. From the list of 420 Italian public hospitals [33], a random stratified sample of 130 hospitals representative of the geographical area and number of beds was extracted, to obtain a sample size of at least 90 participating hospitals; for the 2021 survey, focused also on the risk from hazardous drugs, the sample was driven from the 354 public hospitals including at least one oncology unit.

The interviewers contacted all extracted hospitals. In case of refusal or unavailability, the interviewers contacted the center consecutively present on the list in the same layer.

### 2.3. Target of the Interviews

In 2017, the survey involved Hospital Directors, Health and Safety Managers (Responsabili del Servizio di Prevenzione e Protezione, RSPP), and Heads of Technical and Rehabilitative Nursing Services (SITRA, if present in the hospital), grouped in the category “safety managers” (SM). Nurses from medical, surgical, and infectious diseases units, were interviewed representing the workers exposed to the risk of sharp injuries. In 2021, RSPP and hospital pharmacists were interviewed as safety managers, and oncology nurses were included. It was not possible to interview all the expected professionals in all the hospitals, but at least one representative of nurses and an SM was interviewed from each hospital.

### 2.4. Statistical Analysis

The analyses were systematically produced by comparing the nurses’ and SM’s responses. To compare the proportion observed in 2017 and 2021, a two-proportion z-test was performed. The threshold for significance was set at *p* < 0.05.

Estimates of the studies are subject to a maximum margin of error (*p* < 0.05), respectively, of ±7.3% for the nurses’ target and ±7.9% for the SM target. In order to guarantee the representativeness of the sample, ex post weighting of the results was arranged. Specifically, two weightings were planned, one for the nurses’ sample and one for the SM sample. Both weightings re-aligned the joint distribution (geographic area by hospital size) with the expected percentage distribution of the hospitals. 

A comparison between means was performed using the *t*-test for unpaired data. The threshold for significance was set at *p* < 0.05.

## 3. Results

### 3.1. Sample Characteristics

Participating hospitals were 97 in 2017 and 117 in 2021 (response rate: 75% and 90%, respectively). An infectious diseases department or one or more infectious diseases units were present in 58% and 77% of hospitals, respectively. The sample distribution by geographical area and number of beds is reported in Table 1. 

The first survey included 56 Hospital Directors, 66 RSPP, 13 SITRA managers and 180 nurses (110 working in medical units, including infectious diseases, and 70 in surgery). In the second survey, 65 RSPP, 85 hospital pharmacists, and 150 nurses (100 working in medical units, including infectious diseases, and 50 in oncology) were interviewed. Females represented 70% of nurses and 39% of SM. The median age was 45 years among nurses, and 55 among SM, with no significant differences between the two surveys.

### 3.2. Surveys Results

Knowledge of the requirements of the Legislative Decree 19/02/2014 (implementing the Directive 2010/32/EU)

When asked about which preventive measures were required in the Directive from a list of items also including safety requirements from previous legislation, the proportion of nurses who answered correctly was 70% in 2017 and 58% in 2021 (*p* = 0.00313), and among SM, 80% in 2017 vs. 51% in 2021 (*p* < 0.0001). In both surveys, the best-known measure was the adoption of devices integrating a safety mechanism (nurses, 85% in 2017 and 67% in 2021, *p* = 0.0002; SM, 88% and 57%, *p* < 0.0001). Banning of recapping was the second best-known prevention measure among nurses: 80% in 2017, and 69% in 2021 (*p* = 0.0035), though in 2021 it was considered mandatory by 75%. The least-known measure was the elimination of unnecessary needles (nurses, 59% and 44%, *p* = 0.0097; SM, 74% and 53%, *p* = 0.0069).

Information, Education and Training

In the first survey, nurses (73%) and SM (92%) reported that their unit/hospital organized information activities on biological risks and preventive measures related to the use of sharps for the staff. Information was disseminated through meetings (80%), posters (50%), publication of documents on the hospital website (40%), distribution of brochures or informative materials in the hospital (40–50%), and diffusion by corporate email (20–30%).

In 2017, SM reported that education and training initiatives on biological risks and preventive measures related to the use of sharps had been carried out in their hospital, involving nurses and other healthcare workers from all (83%) or most (10%) of the hospital units; 2% reported no specific initiatives. In 2021, these proportions were 40%, 30%, and 26%, respectively (*p* < 0.001 for all comparisons); in 82% of cases, these subjects were treated within courses on COVID-19 prevention. Regarding newly employed health personnel, sharps safety was included within general courses on the prevention of risks at work (26% in 2017, 39% in 2021), or more frequently addressed in specific courses (65% and 55%, respectively, in the two surveys).

In 2017, 54% of nurses said they had recently attended courses or received specific training on the risks from bloodborne pathogens and the safe use of sharps, for an average of 3.3 days in the past 12 months, mainly through lectures and coaching by colleagues. In 2021, participation over the last two years (2019–2020, to correct for the impact of the COVID-19 pandemic) was reported by 25% (*p* < 0.0001), for an average of 1.7 days, again mainly through lectures, followed by distance learning (Figure 1). In the same period, 41% of RSPP and 30% of pharmacists attended specific courses for an average of 6.3 and 3.3 days, respectively.

Behaviors regarding the use and elimination of sharps

Sixty-eight percent of nurses in 2017, and 66% in 2021, acknowledged they made at least one incorrect maneuver using needles and sharps in their daily work routine, in most cases sporadically. Considering those performed on a routine basis (always or often), most were infrequent (4% of nurses disassembled used devices by hand or manipulated used needles with both hands, 3% passed used sharps directly to colleagues, 1% bent or broke used needles). The most frequent misconduct in both surveys concerned the disposal of used sharps: in 2017, 14% of nurses recapped used needles; 50% placed used safety-engineered devices in temporary containers, such as kidney trays or wraps, or inappropriate containers, such as infectious waste containers, and 34% did so with conventional devices. In 2021, the corresponding figures were 2% (*p* = 0.0003), 22% and 12% (*p* < 0.0001 for both comparisons).

Specifying and implementing safe procedures for using and disposing of sharp medical instruments and contaminated waste, and placing technically safe containers for the handling of disposable sharps as close as possible to the areas where sharps are being used, were investigated in the first survey: established, written safe procedures were available in 98% of cases; nurses reported that sharps containers were available on trolleys (86%), in the patient’s room (10%) or in the nurses’ workstation (4%), as confirmed by SM. In both surveys, nurses reported that these procedures were treated during courses (88% in 2017 and 87% in 2021) among the most important subjects.

Adoption of devices integrating a safety mechanism (safety-engineered devices, SED)

Adoption of SED, as reported by end users or SM, was significant in both surveys (Figure 2), with a vast majority of hospitals having introduced most, or all, investigated SED types (peripheral venous and arterial catheters, blood collection sets with straight or winged-steel needle, lancets, blood gas syringes, hypodermic syringes, IM needles, insulin pens, retractable scalpels). Some SED types were, however, not always routinely available and used (Table 2). A significantly increased availability of phlebotomy sets, IV catheters, lancets, and syringes for subcutaneous injection, and a decreased availability of safety arterial catheters were reported by nurses in 2021. Hospital pharmacists, in comparison to SM, reported in 2021 a lower availability of arterial blood gas syringes, and increased availability of hypodermic needles for intramuscular injection. Conventional devices were totally replaced according to 48% of nurses and 42% of SM in 2017; in 2021, the corresponding figures were 47% and 59%, respectively, and 31% according to hospital pharmacists. When only partial replacement was reported, the average proportion of substitution was around 70%. The two main reasons for partial or no replacement reported by nurses and SM in 2017 were stocks of conventional devices to be consumed, and the cost of SED acknowledged, respectively, by approximately 50% and 40% of respondents in both categories. In 2021, the cost was again reported as the main reason (pharmacists 29%, RSPP 25%); pharmacists also reported that they had not received requests from hospital wards (28%), or that these devices could not be purchased (24%).

Vaccinations

In 2017, only Hepatitis B vaccination among nurses was investigated, with 89% reporting complete immunization. In 2021, nurses reported being vaccinated against Hepatitis B in 89% (96% in infectious diseases), COVID-19 in 97%, and influenza (in the last season) 47% (54% in oncology). The corresponding figures among RSPP were 76%, 98% and 57%, and among hospital pharmacists, 59%, 98%, and 66%. According to nurses, the importance of Hepatitis B vaccination was the least frequently discussed topic during courses on biological risks and preventive measures related to the use of sharps, reported by 68% in 2017, and 46% in 2021.

Injuries, reporting and post-exposure protocol

In 2017, 50 out of 66 RSPP were able to provide the exact number of injuries for their hospital: they reported a mean of 26.1 injuries/year from needlestick or cuts per hospital (median 14, SD 30) (15 for hospitals with fewer than 500 beds, 42 for those over 500 beds); no data were collected on individual injuries. In 2021, 51 out of 65 RSPP reported a mean of 31.3 (median 16, SD 35), (27 < 500 beds, 38 > 500 beds) (two-tailed *p*-value = 0.4251, NS). Two percent of interviewed nurses suffered an injury involving needles or sharps over the last 12 months, and 100% officially notified the injury and activated the post-exposure protocol.

Most nurses (89% and 81% in 2017 and 2021, respectively) and SM (97% and 92%) reported that their hospital had an occupational injury management protocol. When asked about its contents, the main elements recalled by the interviewees were the reporting and recording of the exposure (95%); the reconstruction of the accident including the root cause analysis (85%) and the adoption of corrective actions (65%). Regarding the management of biological risks, the protocol provided the operating instructions for the assessment of the serological status of the source (92%), and the offer of post-exposure prophylaxis to prevent HIV infection (82%) and hepatitis B (76%).

The procedure for the official notification of the injury was considered easy to comply with by 31% of nurses in 2017, and 12% in 2021, while 30% and 40%, respectively, in the two surveys considered it to be objectively complex and an obstacle to reporting.

Obstacles to achieving sharps safety

To conclude, participants were asked to report their overall opinion on the degree of implementation of the legislation for the prevention of sharp injuries in their hospital, and what they thought to be an obstacle to achieving optimal levels of safety in the use of sharps for nursing/health personnel in their hospital (open question).

In 2017 and 2021, respectively, nurses reported complete implementation in 79% and 61%, partial implementation in 20% and 38%, and no implementation in 1%; as regards SM, complete implementation was reported by 75% in 2017 and 68% in 2021, and partial implementation in 25% and 32%, respectively.

The main obstacles perceived by respondents to achieving sharps safety in both surveys were staff shortage and working in conditions of emergency, followed by a lack of financial resources. Stress and fatigue at work, and difficulty handling sharp objects while wearing full personal protective equipment to protect against SARS-CoV-2, were reported as significant issues hampering safety in 2021 (Figure 3).

## 4. Discussion

Italian health professionals from a representative sample of public hospitals reported a good level of implementation of the main elements for the prevention of sharp injuries included in Directive 2010/32/EU.

Regarding the support from the hospital management, specific information, education and training were provided to permanent staff and newly employed workers; safe procedures for the use and disposal of sharp medical instruments and contaminated waste were implemented, and these subjects were extensively treated during courses; the adoption of safety devices was widespread, and incident reporting procedures and post-exposure protocols showed a clear effort to focus on systemic factors rather than individual mistakes, and improve occupational safety by learning from experience. Although in 2021 a general decrease in the level of information was detected, as well as a significant reduction in educational initiatives for HCWs, and maybe also of the possibility for nurses to attend courses, the frequency of incorrect behaviors in the use of sharps, such as recapping and improper disposal of used devices, decreased significantly, and the average injuries per hospital showed a nonsignificant increase. Moreover, nurses showed good compliance with recommended vaccination programs and also with injury reporting, notwithstanding the perceived difficulties in the procedure for the official notification.

Hospital safety climate has been found to be significantly associated with HCWs’ safety. Demonstrable management support for safety programs was significantly associated with a higher HCWs’ compliance with safe work practices (e.g., never recap, proper sharps disposal), and decreased exposure incidents [34], providing a possible explanation for the observed decrease in incorrect behaviors when handling used sharps, and for the relatively stable number of injuries. Conversely, increased compliance was also significantly associated with the absence of job hindrances, such as time constraints and increased workload, while decreased exposure incidents were associated with safety feedback and training. In 2021, nurses reported working in emergency conditions and stress and fatigue among the most important perceived obstacles to achieving sharps safety, and a diminution in the attendance/organization of specific courses over the last two years.

However, the vast majority (89% in 2017 and 79% in 2021) had attended specific courses in recent or previous years, and this fact, in combination with other interventions (information, and established procedures for sharps disposal) has perhaps influenced nurses’ attitude and prevented the further performance of incorrect maneuvers. Together with the implementation of SED, these measures could have prevented an increase in injuries from occurring.

The results of the two surveys are in line with those obtained in SIROH hospitals in 2013 [27], in terms of provision of information and education (also for newly employed workers), the availability of sharps containers and safe procedures for the use and disposal of sharps, and Hepatitis B vaccination rates. The frequency of recapping, not self-reported but measured by examining sharps containers, was higher, with an average percentage of recapped needles of 28%. SIROH data from 1994 to 2016, out of a total of 52,833 injuries with hollow-bore needles and lancets, showed that 3168 of these (6%) had occurred during the recapping of the device, though a significant decrease was observed over time: from 11.2% in 1994 (367/3278) to 2.3% in 2012 (69/3008) [35].

As for SED, the availability in 2013 was lower: the 100 SIROH hospitals had adopted on average four of the available types of device (range 1–11 types, including Huber/Gripper needles for port-a-cath access, thoracentesis catheters and fistula needles for hemodialysis); however, the replacement rates of hollow-bore, blood-filled needles, carrying the greatest risk of infection in case of injury, were higher (IV catheters 91%; blood collection sets 87%) [27]. The sore point is that, in none of these investigations, there was a total replacement of conventional devices with their safety counterpart: in 2013, the SIROH hospitals reported a total replacement in 70% of cases; in 17% it was total for some devices and partial for others, and in 13% only partial for all safety devices [27]. Although the effectiveness of safety devices is variable [36], as reported also in a Cochrane review [37], several studies report positive results regarding the ability of SED to reduce the number of percutaneous accidents [13,38,39]; the contemporary use of conventional, non-safety devices and SED is not beneficial [40,41]. Nurses reported total replacement in nearly 50% of cases in both surveys; in 2021, RSPP reported total replacement almost in 60% of cases, but pharmacists (in charge of purchasing devices) only in 31%. As in other European studies [41], costs were identified as one of the main reasons for not replacing conventional devices, but in 2021, pharmacists and RSPP indicated also difficulties in ordering safety devices. Device purchases are currently made through regional tenders, and safety features may not be included, or not be prioritized, within the required characteristics of the device, depending on the choices of the committee developing the tender. The lack of requests from the departments or units (28% according to pharmacists, and 10% according to RSPP) is a less understandable reason: the choices in terms of safety must be made by the hospital management, and certainly shared with the end users, but they must not be left to the latter alone.

As stated above, the COVID-19 epidemic has had an impact on some aspects: the share of nurses who recently attended courses on sharps safety more than halved compared to the previous survey, and many of these courses employed distance learning as the preferred methodology. Among the safety barriers reported, the perception of working in conditions of emergency almost doubled, and new issues included stress and fatigue at work and difficulties in handling sharps resulting from full protective clothing. Moreover, the average number of sharp injuries increased by 23% when compared to 2017, though this increase was not statistically significant, and the study design did not allow further evaluations, including changes in their characteristics that might help in designing further preventive strategies.

These figures must be taken cautiously, as there are no annual data on occupational injuries with sharps at a national level in Italy in this period. SIROH hospitals, certainly forerunners in terms of applying measures for the prevention of sharp injuries, recorded a decrease in the average number of injuries per hospital in the years following the issue of the Directive, from 64 in 2010 to 29 in 2013, to 20 in 2016 [35]. The recent survey commissioned by EBN, which included 80 of the largest European hospitals in Spain, France, Germany, Poland, and Italy covering over 300,000 healthcare workers, found that the number of sharp injuries had increased significantly in 2020, with an average reported increase of 23%, ranging from +9% in Italy to +32% in Germany. The increased pressure and stress due to COVID-19 were considered the basis of this increment (98% of responses), followed by a lack of safety devices (average 47%, ranging from 31% in Italy to 75% in Poland) and a lack of personal protective equipment (average 45%, ranging from 8% in Italy to 100% in Poland) [30]. Therefore, Italian healthcare workers reported to be better equipped and experienced a less significant increase in occupational injuries than their colleagues in other European countries. Nonetheless, the Italian National Healthcare System (NHS) was heavily stressed by COVID-19: the price that the NHS had to pay both in terms of the number of positive virus cases and deaths among the HCWs was significant and represented a peculiarity compared to what happened in other countries [42]. Considering the increase in the intensity of care due to the selective hospitalization of patients with COVID-19 and other emergency medical or surgical conditions in 2020 and 2021 [43], the low rate of staff turnover (due to the shortage of collaborators) [42], the consequent pressure to immediately recruit new nursing graduates [44], as well as residents and even medical students for some roles, the system has shown a good level of resilience and the implementation of occupational accident prevention measures prior to this emergency has certainly helped protect healthcare workers during the pandemic, although other coping strategies may have intervened. A study on organizational resilience processes among HCWs in Switzerland in the first year of the pandemic found that the difficulties in implementing COVID guidelines and protection measures fell in the category of problematic situations associated with the development of new standards, that is, HCWs became used to new ways of working and developed new skills to perform effectively [45]. A retrospective study performed at Sisli Hamidiye Etfal Training and Research Hospital in Istanbul, Turkey, found that sharp injuries rates per 1000 healthcare workers decreased during the pandemic period, mainly due to the procurement of personal protective equipment and organizing well-designed training and awareness programs at the beginning and during the COVID-19 pandemic period, even if the possibility of underreporting may limit the significance of the results [46].

Underreporting should always be considered when evaluating sharp injuries. A recent study in Portugal detected a 45% underreporting of percutaneous injuries, whose main reasons were the underestimation of transmission risk (49%) and bureaucracy (41%) [47]. Nurses in the two surveys reported increasing difficulties in the official notification of occupational injuries; therefore, underreporting may have affected our results as well. However, it is encouraging that nurses who had experienced a recent injury acknowledged having formally reported it and activated the post-exposure protocol.

While these findings are subject to several limitations related to the study design, and especially to recall and desirability bias, the two surveys were conducted anonymously by an external provider, to increase participation and decrease the fear of negative judgments and possible administrative sanctions in case of non-compliance with legal requirements. Furthermore, the results of the two surveys are consistent with each other and offer a multifaceted picture with positive and negative elements, most likely in line with the working reality of Italian public hospitals. Future studies should employ a cohort design targeting specific occupational groups and compare these groups across countries. To assess the impact of Directive 2010/32/EU in the 28 Member States, surveillance at the EU level should be implemented through the development of a permanent observatory to provide detailed and up-to-date information and data on sharp injuries and to further assess the role of contributing and preventive factors.

## 5. Conclusions

Since the introduction of the European legislation for the prevention of sharp injuries, there have been significant improvements in Italy regarding the safety of health professionals from the risk of bloodborne infections, both from the point of view of the behavior of the HCWs and of the preventive interventions implemented by employers. Although average injuries per hospital did not show a decrease, the safety measures implemented in recent years could have helped protect healthcare workers during the pandemic, mitigating the potential impact of the pandemic on the increase in situations at risk of injury for healthcare workers, engaged in all specialties to provide intense and difficult assistance to patients. As new risks increasingly emerge that could lead to significant changes in healthcare activities, research and practical efforts should be focused on providing and evaluating the effectiveness of multi-intervention preventive packages on HCWs’ safety and the risk of sharp injuries. Some aspects deserve attention, such as training in safer behaviors, the availability of devices integrating a protection mechanism, and the facilitation of procedures for the official notification of at-risk injuries, all measures that reduce the likelihood of infection with bloodborne pathogens in health workers.

## Figures and Tables

**Figure 1 ijerph-19-11144-f001:**
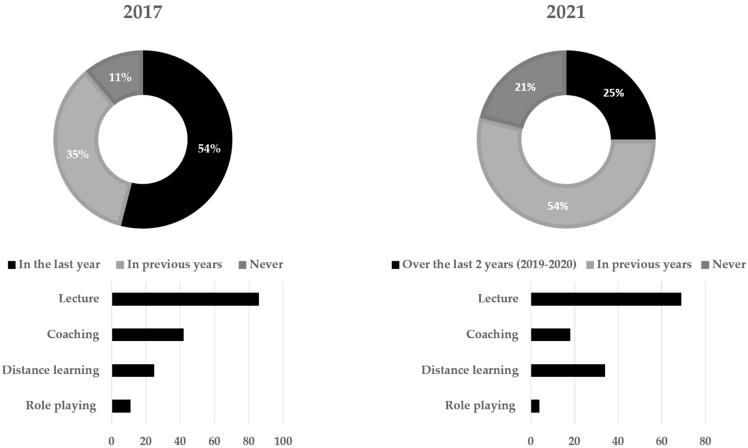
Nurses’ participation in training activities on biological risk and preventive measures related to the use of sharps, Italian Observatory on Needle and Sharps Safety, 2017 vs. 2021.

**Figure 2 ijerph-19-11144-f002:**
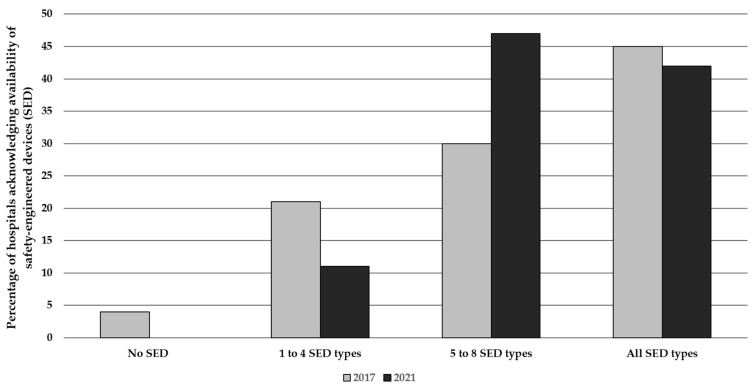
Availability of Safety-Engineered Devices (SED) in a sample of Italian hospitals representative by geographical area and size, 2017 (97 hospitals) vs. 2021 (117 hospitals). *p* = NS for all comparisons.

**Figure 3 ijerph-19-11144-f003:**
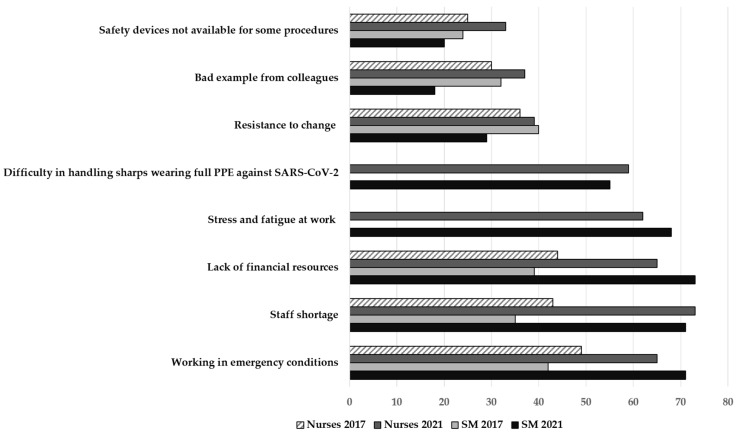
Percentage distribution of respondents’ opinions regarding obstacles to achieving sharps safety, 2017 vs. 2021. SM: Safety Managers.

**Table 1 ijerph-19-11144-t001:** Characteristics of the sample of hospitals participating in the Italian Observatory on Needle and Sharps Safety, 2017 vs. 2021.

Geographical Area	200 Beds		201–500 Beds		>500 Beds		Total	
	2017	2021	2017	2021	2017	2021	2017	2021
North-West	4	6	8	15	10	11	22	32
North-East	7	1	4	9	4	10	15	20
Center	8	5	11	11	5	10	24	26
South-Islands	9	4	16	15	11	20	36	39
Total	28	16	39	50	30	51	97	117

**Table 2 ijerph-19-11144-t002:** Hospital-wide, partial, or no availability, of safety-engineered devices in a representative sample of Italian public hospitals, 2017 vs. 2021.

Safety-Engineered Devices	Nurses					Safety Managers		Hospital Pharmacists		
	2017		2021			2017		2021		
	180		150			135		85		
	N	%	N	%	*p*-Value	N	%	N	%	*p*-Value
**Vacuum-tube phlebotomy sets**										
Routinely available and used	128	71	134	89	**0.0001**	86	64	63	74	0.1441
Not routinely available	36	20	12	8	0.0035	36	27	15	18	0.1677
Not available in the hospital	16	9	4	3	0.0334	13	10	7	8	0.9128
**Peripheral IV catheters**										
Routinely available and used	121	67	125	83	**0.0013**	86	64	60	71	0.3650
Not routinely available	43	24	18	12	0.0086	33	24	20	24	1.0000
Not available in the hospital	16	9	7	5	0.1996	16	12	5	6	0.2181
**Arterial Blood Sampling syringes**										
Routinely available and used	101	56	77	51	0.4496	72	53	29	34	***0.0081*** *
Not routinely available	34	19	47	31	0.0129	37	27	33	39	0.1049
Not available in the hospital	45	25	26	17	0.1204	26	19	23	27	0.2351
**Arterial catheters**										
Routinely available and used	74	41	41	27	***0.0124*** *	76	56	24	28	***0.0001*** *
Not routinely available	30	17	55	37	0.0001	32	24	36	42	0.0057
Not available in the hospital	76	42	54	36	0.2989	27	20	25	29	0.1507
**Lancets**										
Routinely available and used	99	55	114	76	**0.0001**	63	47	39	46	1.0000
Not routinely available	30	17	26	17	0.9893	29	21	29	34	0.0556
Not available in the hospital	51	28	10	7	0.0000	43	32	17	20	0.0773
**Syringes for subcutaneous injection**										
Routinely available and used	86	48	92	61	**0.0188**	49	36	47	55	**0.0086**
Not routinely available	38	21	30	20	0.9110	36	27	18	21	0.4470
Not available in the hospital	56	31	28	19	0.0140	50	37	20	24	0.0517
**Hypodermic needles for IM injection**										
Routinely available and used	76	42	77	51	0.1231	53	39	58	68	**0.0001**
Not routinely available	39	22	42	28	0.2291	32	24	15	18	0.3690
Not available in the hospital	65	36	31	21	0.0031	50	37	12	14	0.0004
**Insulin pen needles**										
Routinely available and used	95	53	81	54	0.9118	68	50	39	46	0.6101
Not routinely available	35	19	41	27	0.1179	33	24	27	32	0.3022
Not available in the hospital	50	28	28	19	0.0703	34	25	19	22	0.7517
**Scalpels** §										
Routinely available and used	74	41	51	34	0.2255	65	48	27	32	***0.0239*** *
Not routinely available	25	14	54	36	0.0000	25	19	33	39	0.0015
Not available in the hospital	81	45	45	30	0.0074	45	33	25	29	0.6459

Statistically significant differences are indicated in bold; in bold italics, statistically significant decreases. * availability significantly decreased. § in 2021, nurses from surgery were not included.

## Data Availability

Aggregated data used in this study are available from the corresponding author on reasonable request.

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
