# Peer review of "Prevention from Sharp Injuries in the Hospital Sector: An Italian National Observatory on the Implementation of the Council Directive 2010/32/EU before and during the COVID-19 Pandemic"

_ijerph, 2022, doi:10.3390/ijerph191711144_

Round 1
Reviewer 1 Report
The authors have used stratified random sampling method to show the impact of implementation of Directive 2010/32/EU for the prevention from sharp injuries in Italian hospitals, safety managers and nurses from a representative sample of public hospitals. The respondents were interviewed in 2017 and in 2021 regarding required preventive measures. Participating hospitals were 97 and 117, respectively; overall, 285 safety managers and 30 330 nurses were interviewed. Using simple percentage change they found that the intervention was effective. They have not used any sophisticated statistical techniques like DID method ITS method or ARIMA method which are prominently used for this kind of intervention analysis. DOI https://doi.org/10.2147/RMHP.S275831
But the topic is ok.
Author Response
We thank the Reviewer for the suggestions, and the useful article, regarding the statistical analysis. Unfortunately, the suggested methods could not be applied to our data, as we did not have a pre- and post- intervention population; considering the implementation of the Sharps Directive as the intervention, both surveys were carried out in the post-intervention period; and data from the 2013 survey were not suitable as a pre-intervention group.
As we compared the level of implementation of the different elements of the Sharps Directive in terms of proportion of answers, we included two proportion z-test when comparing proportion obtained in 2017 and 2021, in the text and in table 2.
Reviewer 2 Report
The submitted manuscript contains interesting data and is undoubtedly a good account of the situation, but I have serious doubts about its scientific soundness.
The manuscript lacks a theoretical framework to support the analysis of the variables that determine risk prevention, neither in general, nor specifically with respect to sharp injures.
The results of two interviews conducted at different points in time are compared. The structure of each of the interviews is not described, nor whether they were identical for all interviewees. Both interviews are different and it is not clear how many common questions are analysed or if open questions were included and which ones were included.
No information is provided on any ethical committee that authorised the research, nor on the informed consent of the participants.
Results are presented as percentages, without significance analysis of differences found between years and without analysis of relationships between variables. These analyses mark a clear difference between a research study and a report.
The discussion, limited by the absence of a theoretical framework, nevertheless includes references not previously presented.
Author Response
We presented the results of two nationwide surveys, involving a representative sample of hospitals, aimed at evaluating the level of implementation of the preventive measures enforced in Italy through European and national legislation, assessed before and during the COVID pandemic. These measures are based on epidemiological evidence deriving from studies and surveillance programs of occupational injuries and infections in healthcare workers. We included these evidences in the Introduction section, which was rewritten to provide a sufficient background and include more relevant references, as suggested.
In the Methods section, line 159-173, we described in more detail the structure of each of the interviews, reporting the differences between the two questionnaires, the common questions analyzed and the only open question included in the comparisons.
Information is now provided on ethical committee clearance and on the informed consent of the participants (line 179-184, and Institutional Review Board and Informed Consent Statement)
The significance analysis of differences found between years was included in the Results.
The discussion was improved to better describe and compare the results, and support the conclusions.
Round 2
Reviewer 1 Report
OK
Author Response
We further acknowledged the limits of the study design, better explaining the study aim, in the Introduction and Discussion sections.
Reviewer 2 Report
The authors have made modifications to clarify the methodological aspects. The context in which the study takes place is better described. However, the authors still do not provide a theoretical framework to support the study, which remains merely descriptive of the situation.
Author Response
We thank the reviewer for providing important insights.
We rewrote the Introduction, adding general and specific background and relevant references and better explaining the background and purpose of the study: we hope to have provided a theorethical framework to support the study. Moreover, we modified the Discussion and rewrote the Conclusions; the abstract was modified accordigly.